# Maternal Gut Dysbiosis Alters Offspring Microbiota and Social Interactions

**DOI:** 10.3390/microorganisms9081742

**Published:** 2021-08-15

**Authors:** Zihan Zhang, Chao Xue, Mengyao Ju, Jiawei Guo, Minghui Wang, Sijie Yi, Xianfeng Yi

**Affiliations:** 1College of Life Sciences, Qufu Normal University, Qufu 273165, China; zzh20001205@163.com (Z.Z.); xuebhc1001@163.com (C.X.); jmy17853728901@163.com (M.J.); 18963523133@163.com (J.G.); wmh19952021@163.com (M.W.); 2College of Life and Environmental Sciences, University of Exeter, Exeter EX4 4QD, UK

**Keywords:** maternal gut dysbiosis, antibiotics, offspring, gut microbiota, metabolome, social behavior

## Abstract

Increasing application of antibiotics changes the gut microbiota composition, leading to dysbiosis of the gut microbiota. Although growing evidence suggests the potential role of gut dysbiosis as the cause of neurodevelopmental disorders and behavioral defects, a broad gap of knowledge remains to be narrowed to better understand the exact mechanisms by which maternal gut dysbiosis alters microbiota development and social interactions of offspring. Here, we showed that maternal gut dysbiosis during gestation is a critical determinant of gut microbiota and social interactions off mouse offspring. Gut microbiota of 2-week-old offspring showed significant changes in response to maternal antibiotic treatment. We even detected distinct effects of maternal oral antibiotics on gut microbiota of 14-week-old offspring. Compared to controls, offspring born to antibiotics-treated mothers displayed reduction in sociability and preference for social novelty, suggesting that the altered offspring social behavior was closely linked to dysbiosis of maternal gut microbiota. Our study opens the possibility to better understand the mechanism of how maternal gut microbiota vertically impairs social interactions of offspring in animal models, providing support to the maternal gut microbiota as a potential mediator between offspring microbiota and behaviors.

## 1. Introduction

Antibiotics were introduced into the field of infection therapy less than 100 years ago. Over the past several decades, increasing application of antibiotics has drastically changed the course of healthcare and humanity [1]. Apart from being used for treating and preventing infectious diseases, low dose, unlabeled broad-spectrum antibiotics have been widely used for animal farming and as well as for agricultural purposes. It has been estimated that over half of the antibiotics manufactured in the U.S. are used for agricultural purposes, contributing negative impacts to the healthcare of humanity. Despite its necessity and efficiency to eradicate pathogens in hosts in some cases, antibiotics not only change the gut microbiota diversity but may also partially diminish naturally occurring commensal bacteria contained in intestines, leading to gut microbiota dysbiosis with lasting consequences [2,3].

It has been suggested that microbiota in the early stage of life shows a profound influence on host development and the functions of organisms [4]. When an individual is born, indigenous microbes have been found to rapidly and densely habituate and occupy the intestines of the newborn babies. The maternal gut microbiota is a leading force during pregnancy or breastfeeding in defining offspring gut microbiota that are subjected to changes due to delivery manners and maternal effects [5]. During lactation, maternal antibiotics will continue to alter the neonatal microbiome, exerting lasting consequences on offspring [6,7,8]. Over the last few decades, different studies on humans and mouse models have revealed strong associations between antibiotics-induced gut microbiota dysbiosis and host diseases, neuroinflammatory and psychiatric disorders as well as brain development [9,10,11,12,13]. A recent review by Dinan et al. [14] has shed light on the influences of gut microbiota dysbiosis in mediating neurological disorders (e.g., autism) and behavior such as anxiety, stress, etc., [15,16], which has been supposed to directly or indirectly affect social behaviors.

During the last decades, the microbiota–gut–brain axis has become a hotspot in modern science elucidating the potential mechanism of how the gut microbiota modulates brain function as well as social behaviors [17]. It is well accepted that social behavior plays an important role in the survival and prosperity of animals; however, neurological disorders induced by gut dysbiosis have been claimed to result in disruption of normal social behaviors. Despite growing evidence which has suggested a potential role of gut dysbiosis to cause mood and behavioral defects, a significant gap of knowledge needs to be narrowed to understand the exact mechanisms by which maternal gut dysbiosis affects microbiota development and social interactions of mouse offspring. Although some investigations have verified altered gut microbiota composition of patients suffered from neurodevelopmental disorders (e.g., autism), a deep understanding of the fundamental mechanisms by which gut microbiota in animal’s early life mediates the development and occurrence of social behaviors is still very limited [18]. In addition, few studies have been conducted to understand the sociability and preference for social novelty in response to early life gut dysbiosis. Moreover, it remains unclear how long the effects of maternal gut dysbiosis will last. Here, using low dose oral administration of a combination of vancomycin and neomycinsulphate, which have low oral bioavailability, we showed that maternal gut dysbiosis during gestation is a critical determinant of gut microbiota and social interactions of mouse offspring.

## 2. Materials and Methods

### 2.1. Ethical Approval Statement

We performed the experiments according to the guidelines issued by the Institutional Animal Care and Use Committee at Qufu Normal University and followed the Regulations for Experimental Animals issued by China’s State Council in January 1988. The animal use protocol in this study has been approved by the Animal Ethical and Welfare Committee of Qufu Normal University, Approval No. QFNU 2021020.

### 2.2. Mice and Antibiotics Administration

Adult KM (Kunming) mice (8 weeks of age) from the same litter were obtained from Jinan Pengyue Experimental Animal Breeding Co., Ltd., (Shandong, China), and were placed in a 25 °C room on a 12 h light/dark cycle. After acclimation of 1 week, the female mice were mated by housing an adult male and two females in a single cage. The male was then removed after 5 days’ mating. By checking postcopulatory plugs to confirm pregnancy, female mice selected for the treatment group were administered with broad-spectrum antibiotics vancomycin (1 mg/mL) and neomycinsulphate (5 mg/mL) (Shanghai Aladdin Biochemical Technology Co., Ltd., Shanghai, China) dissolved in drinking water during the whole period of gestation (for 20 days). Meanwhile, regular drinking water instead of vancomycin and neomycinsulphate was provided to the control pregnant females. Pregnant mice were housed individually and provided with regular drinking water and standard rat chow (4% fat, 20% protein, 70% carbohydrate; Shenyang Maohua Biotechnology Co., Ltd., Liaoning, China) ad libitum under room temperature. Pups weaned at 2 weeks of age were separated from their mothers. Cecal samples were collected in mothers for microbiome analysis after the three-chamber paradigm test for sociability and preference for social novelty. Ten pups from antibiotics-treated group and eight pups from control group were randomly selected and sacrificed immediately to collect cecal samples for microbiome analysis. The remaining pups of both groups were housed individually and received regular drinking water and standard rat chow *ad libitum* till 14 weeks for cecal sample collection and three-chamber paradigm test.

### 2.3. Test of Sociability and Preference for Social Novelty

In this study, we used the three-chamber social test apparatus to measure sociability and preference for social novelty of mice following the previous protocol [19]. Sociability of the test animals in our study was defined as propensity to interact with an unfamiliar mouse in one chamber rather than with an identical empty chamber. Here, preference for social novelty of the test animals was regarded as propensity to interact with a naive mouse that has never been met, as compared to a previously encountered mouse [20]. The apparatus used in this study was made of polymethyl methacrylate box (length × width × height: 60 cm × 30 cm × 60 cm) with partitions that separate the box into three identical chambers [21,22]. The doors on the partitions, when opened, allowed the test animal to move between chambers freely [20]. At the phase I for habituation, the test animal was placed in the center of the apparatus which allowed it to move freely in all three chambers for five minutes. After the 5 min habituation phase, the test animal was confined in the center chamber by closing the two doors. Then, an unfamiliar adult female (stranger 1) was put inside a cylinder wire cage centered in one of chambers placed at the two sides for sociability test (phase II). At the same time, we placed an identical empty wire cage in the center of the opposite chamber. The doors were then re-opened, which allowed free movement of the test animal on the bottom of the test apparatus for another 5 min. The test animal was allowed to initiate social contact with the stranger through the evenly-distributed bars along the peripheral region of the cage, while the stranger mouse was unable to initiate any social contact. Nose contact was allowed but aggressive interactions were prevented by the wire cage. Therefore, this deployment allowed us to monitor whether the test animal initiated social interaction. To avoid chamber bias, locations of the empty wire cage and the stranger mouse were randomly exchanged between the two chambers for the test animals. Measures of entries between chambers, travel distance in each chamber, and time spent sniffing each wire cage containing the unfamiliar mouse and the empty cage on the opposite side of the apparatus were taken for 5 min using Any-maze video tracking system from Stoelting Co. (version 6.0, Wood Dale, IL, USA).

Immediately after the test for sociability, we did another 5 min test to measure preference for social novelty (phase III). The original stranger mouse (i.e., stranger 1) was confined in its wire cage placed on one side of the apparatus. Then, a naive unfamiliar mouse (i.e., stranger 2) was put in the wire cage on the opposite side, which was not occupied during the sociability test. We ensured that the two stranger animals had never met the test animal or each other. Again, we changed the locations of the stranger 1 and stranger 2 randomly between the two chambers to avoid any chamber bias. Identical measures were recorded as previously described in the sociability test. In our study, antibiotics-treated mothers and controls were tested individually after weaning, while offspring born to the antibiotics-treated mothers and control offspring were tested at 14 weeks old.

### 2.4. DNA Extraction

Total genomic DNA of cecal samples was extracted in OE Biotech Company (Shanghai, China) using a QIAamp DNA Stool Mini Kit (Qiagen, Germantown, MD, USA) following the instructions of manufacturer. Concentration and purity of DNA were qualified with NanoDrop and agarose gel. We used the barcoded primers and Tks Gflex DNA Polymerase (Takara) to run PCR amplification with the genome DNA being the template. V3-V4 variable regions of 16S rRNA genes was amplified with adaptors-linked universal primers 343 F and 798 R.

### 2.5. Bioinformatic Analysis

Raw sequencing data, which were in FASTQ format, were pretreated to detect and then remove the ambiguous bases (N) using Trimmomatic software. In addition, we used sliding window trimming approach to remove low-quality sequences with an average quality score below 20. We then used FLASH software to assemble paired-end reads after trimming [23]. Several parameters, 10 bp of minimal overlapping, 200 bp of maximum overlapping and 20% of maximum mismatch rate, were selected for assembly. Reads with ambiguous, homologous sequences or below 200 bp were discarded to further denoise the sequences. We retained reads with 75% of bases above Q20 but removed those with chimera by using QIIME software (version 1.8.0) [24].

The primer sequences of clean reads were removed and then clustered using Vsearch software to generate operational taxonomic units (OTUs) with 97% similarity cutoff [25]. An OTU table including the number of sequences per OTU in each sample has been constructed. QIIME package was used to select the representative read of each OTU. Using RDP classifier (confidence threshold was 70%) [26], we annotated and blasted all representative reads against Silva database Version 123 (16s rDNA).

### 2.6. Data Analysis

The rarified OTU table were summarized in QIIME [27], to see the effects of antibiotics treatment on the community richness and community diversity of cecal samples. Rarefaction curves were calculated and differences in α-diversity were analyzed at 55,010 sequences/sample. The OTU richness was estimated by the calculated Chao1, Observed Species, Shannon, Simpson indices and Faith’s PD based on the total number of OTUs. One-way ANOVA with Tukey’s test was used to identify differences in the alpha diversity of the gut microbiota using community richness (e.g., Shannon index, Simpson index, observed-species, good_coverage, Chao 1 index and Faith’s PD) between the antibiotics-treated animals and controls [28].

The relative abundances of microbial genera were normalized using a variance stabilizing transformation of arcsin (abundance 0.5) [29]. The analysis of β-diversity (between-sample diversity) was performed by calculating both the Bray–Curtis dissimilarity and the weighted UniFrac distance in QIIME. In this study, we used principal coordinate analysis (PCoA) and distance matrices to analyze the bacterial community data of mice with PRIMER 7 software. Molecular variance (Adonis) was analyzed to test the differences in gut microbiota compositions. The statistical significance was set as *p* < 0.05 [30]. The relative abundant values of OTUs between antibiotics treatments were detected using algorithm of Linear Discriminant Analysis Effect Size (LEfSe). However, some OTUs might be cut off in accordance with ranking of Kruskal–Wallis test (*p* < 0.05 and the score of absolute log 10 LDA). GraphPad Prism 9 was used to plot α-diversity results and gut microbiota at phylum and genus levels. General linear model (GLM) was used to compare the differences in gut microbiota at phylum and genus levels between different treatment groups, and least significant difference (LSD) was used for multiple comparisons.

## 3. Results

We showed that maternal oral antibiotics administration had significant effects on maternal gut microbiota. Gut microbiota α-diversity was decreased in antibiotics-treated mothers compared to controls (Figure 1). PCoA of β-diversity (by Bray–Curtis dissimilarity and unweighted UniFrac) demonstrated that antibiotics-treated mothers clustered separately from controls (Figure 2a: Adonis, R^2^ = 0.3973, *p* = 0.001; Figure 2b: R^2^ = 0.4071, *p* = 0.001). Gut microbiota of 2-week-old offspring also showed distinct changes associated with maternal antibiotic treatment. Significantly reduced α-diversity was observed in offspring born to mothers treated with antibiotics compared to those born to controls (Figure 1. Moreover, PCoA analysis of β-diversity by unweighted UniFrac distance matrix showed different microbial communities between offspring born to mothers treated with antibiotics and born to controls (Figure 2b; Adonis, R^2^ = 0.2325, *p* = 0.001). We even detected distinct effects of maternal oral antibiotics on gut microbiota of 14-week-old offspring. Gut microbiota α-diversity (Chao 1, good_coverage and Faith’s PD) was decreased in 14-week-old offspring born to antibiotics-treated mothers compared to controls (Figure 3a–c). Moreover, PCoA analysis of β-diversity by unweighted UniFrac showed that gut microbiota of 14-week-old offspring presented distinct changes associated with maternal antibiotic treatment (Figure 3d; Adonis, R^2^ = 0.1282, *p* = 0.001).

Four dominant phyla, including Firmicutes, Proteobacteria, Epsilonbacteraeota and Bacteroidetes, were evaluated to compare the differences in gut microbiota between different groups at the phylum level. Antibiotics-treated mothers had higher relative abundance of Proteobacteria versus controls (*p* < 0.01; Appendix A). However, the relative abundance of the phylum Firmicutes and Epsilonbacteraeota was decreased in antibiotics-treated mothers compared to controls (*p* < 0.001 and *p* < 0.05, respectively; Appendix A). This pattern was also evident in 2-week-old offspring from antibiotics-treated mothers that exhibited significantly decreased relative abundance of Firmicutes, Epsilonbacteraeota and Bacteroidetes in their cecal contents as compared to controls (*p* < 0.001, *p* < 0.001, and *p* < 0.05, respectively). Similarly, Proteobacteria was significantly increased in 2-week-old offspring born to antibiotics-treated mothers compared to control (*p *< 0.001). At the genus level, compared to control mothers, antibiotics-treated mothers had lower relative abundance of *Helicobacter*, *Lachnospiraceae*_NK4A136_group, *Ruminiclostridium 9*, and *Intestinimonas* (*p* < 0.001, *p* < 0.05, *p* < 0.001, and *p* < 0.01, respectively; Appendix A), but higher relative abundance of *Klebsiella* and *Enterobacter* (all *p* < 0.05; Appendix A). Differentially abundant species at the phylum, class, order, family, and genus level in Cladogram between offspring born to antibiotics-treated mothers and controls were also reflected by LEfSe analysis (Figure 4a,b).

In the test sessions for sociability of offspring born to control mothers (Phase II), the number of entries into the chambers containing stranger 1 was not significantly more than the number of entries into the chambers containing empty cage (t = −1.295, df = 10, *p* = 0.225; Figure 5d). This pattern was well reflected in the time to sniff the wire cage (t = −2.117, df = 10, *p* = 0.063; Figure 5e) or travel distances in the chambers (t = 0.314, df = 10, *p* = 0.076; Figure 5f). In the test of preference for social novelty of offspring born to control mothers (Phase III), the number of entries into the chambers containing stranger 2 was not significantly more than the number of entries into the chambers containing stranger 1 (t = 1.771, df = 10, *p* = 0.107; Figure 5d). Amount of time spent to sniff the wire cage containing stranger 2 was significantly more than time sniffing the wire cage containing stranger 1 (t = 5.060, df = 10, *p* < 0.001; Figure 5e). In addition, distances traveled in the chambers containing stranger 2 were significantly more than distances in the chambers containing stranger 1 (t = −3.511, df = 10, *p* = 0.006; Figure 5f). However, 14-week-old offspring born to antibiotics-treated mothers exhibited no preference either to the chambers containing stranger 1 over the chambers containing empty cage in the test for sociability at phase II (t = 1.569, df = 12, *p* = 0.143; t = 1.268, df = 12, *p* = 0.229; t = 1.007, df = 12, *p* = 0.335; Figure 5j–l), or to the chambers containing stranger 2 over the chambers containing stranger 1 in the test of preference for social novelty at phase III in terms of entry times, time spent and travel distance in the test sessions (t = −2.059, df = 12, *p* = 0.062; t = −2.042, df = 12, *p* = 0.062; t = 0.593, df = 12, *p* = 0.564; Figure 5j–l). Despite significant changes in gut microbiota induced by antibiotics (Figure 3), antibiotics-treated mothers (Figure 6a–c) exhibited similar sociability and preference for social novelty to control (Figure 6d–f), as seen from the entry times, the time spent sniffing, and the travel distances in each chamber at phase I, II, and III.

## 4. Discussion

It is widely accepted that maternal gut microbiota is one of the major factors determining gut microbial composition of offspring not only by placenta and amniotic fluid during gestation in utero, but may also by delivery and breastmilk microbiota [31,32,33]. Overall, our results revealed that maternal gut microbiota dysbiosis significantly altered offspring gut microbiota, which is consistent with the notion that pregnant and lactating mothers can shape the microbial communities of their offspring [34,35,36]. However, offspring communities failed to cluster closest to their own mothers’ communities in the PCoA analysis in our study although it is proposed that gut microbiota is established during early life mainly through vertical transmission from mother to offspring [37]. Further evidence was derived from the low percentage of OTUs shared by mothers and their corresponding offspring. This discrepancy may reflect the role of environments in shaping gut microbiota of offspring apart from maternal vertical effects. Consistent with the results in previous studies [38,39], we showed that the antibiotics-treated mothers had lower relative abundance of Firmicutes and Epsilonbacteraeota versus controls. Whereas, the relative abundance of Proteobacteria was increased in antibiotics-treated mothers compared to controls. Oral administration of the broad-spectrum antibiotic combination of low dose vancomycin and neomycinsulphate markedly increased colonization of gut with Proteobacteria, providing supports to the notion that Proteobacterial load is a potential diagnostic criterion for gut microbiota dysbiosis [40]. Previous study has evidenced that the microbiota develops and reaches stability within 3 to 4 weeks of life in mice [41]. Differences in gut microbiota composition between 14-week- old offspring born to antibiotics-treated mothers and controls further suggest that maternal gut microbiota will, beyond our previous thoughts, persist throughout the offspring’s life [42,43].

The three-chamber paradigm test indicated significant differences in sociability and preference for social novelty between offspring born to antibiotics-treated mothers and controls. Offspring born to antibiotics-treated mothers exhibited reduced sociability and preference for social novelty compared to controls, suggesting that gut microbiota plays an essential role in maintaining and modulating mouse social behaviors [38,44]. The data we presented here clearly showed that antibiotics-induced maternal gut microbiota dysbiosis strikingly and vertically modified gut bacterial composition and then impaired sociability and preference for social novelty in the mouse offspring, which resembles social cognition deficits associated with neurodevelopmental disorders [45,46,47]. These observations provided us with a mixed feeling about the role of maternal gut microbiota dysbiosis in controlling social behavior of offspring. However, perturbations of gut microbiota of adult mothers failed to significantly impair their sociability and preference for social novelty. These observations suggest that alterations in the gut microbiota during the critical early developmental periods have been associated with changed behavioral phenotypes in our mice and other mouse models [48,49,50].

We provided some evidence on the correlation between perturbations of gut microbiota and behavioral phenotypes. Interestingly, we found that the relative abundance of Lachnospiracea and Prevotellaceae, which are generally linked to the production of butyric acid [51], as well as Ruminococcus, Ruminococcaceae, Prevotellaceae, Bacilli, and Lactobacillaceae exhibiting probiotic properties [52,53,54,55], were significantly decreased in the offspring born to antibiotics-treated mothers that exhibited reduced sociability and preference for social novelty. However, the relative abundance of inflammation-associated microbiota Enterobacteriaceae, Pasteurellaceae, and Gammaproteobacteria [56,57,58], was decreased in the offspring born to control mothers exhibiting normal sociability and preference for social novelty. Gammaproteobacteria has been proved to contain many common Gram-negative ‘pathogens’ and may involve in producing molecular components that are responsible for cyclical inflammation [59]. Singhal et al. [60] have provided evidence that inflammation is highly associated with depressive- and anxiety-like behaviors. Thus, it is likely that decrease in probiotics but increase in pathogen-like bacteria may result from the disrupted mucosal barrier function and immunity in the offspring born to antibiotics-treated mothers [39]. Gut microbiota dysbiosis has been found to be closely connected to psychiatric and neurological malfunctions, e.g., autism spectrum disorder (ASD), anxiety and depression, Alzheimer’s (AD) and Parkinson’s disease (PD) in humans and mouse models [7,11,20,44]. Therefore, it can be expected that maternal gut microbiota dysbiosis during gestation may represent a mechanism by which the sociability and preference for social novelty of the offspring mice were reduced [45,61,62].

## 5. Conclusions

In summary, our results in the present study showed that dysbiosis of maternal gut microbiota via oral supplementation antibiotics significantly changed maternal microbial composition. More importantly, the composition of offspring intestinal microbiota was significantly modified by dysbiosis of maternal gut microbiota, and further reduced their sociability and preference for social novelty. These findings suggest that maternal gut microbiota dysbiosis can negatively cause a significant shift in gut microbiota of offspring, and then reduce their social interactions. Therefore, our study may provide the possibility to further understand the mechanism of how maternal gut microbiota vertically impairs social interactions of offspring in animal models. Our results also provide support for the maternal gut microbiome as a possible mediator between offspring microbiota and behaviors. These findings provide fodder for further work to explore the importance of the gut microbiome in these relationships. Further microbiota work should be undertaken in an effort to better understand the complex interactions between maternal gut microbiota and offspring behavior.

## Figures and Tables

**Figure 1 microorganisms-09-01742-f001:**
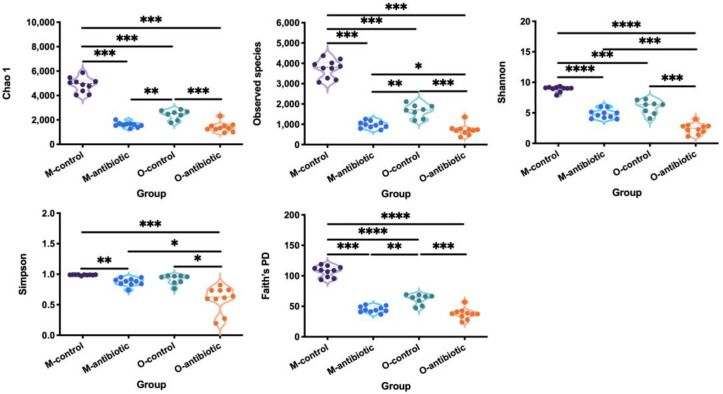
Comparison of the α-diversity indices (Chao 1, the number of species, Shannon, Simpson and Faith’s PD) of mothers and offspring treated with oral antibiotics. M-control and M-antibiotic stand for control mothers and antibiotics-treated mothers, respectively, while O-control and O-antibiotic represent offspring born to control and antibiotics-treated mothers, respectively. An ANOVA with Tukey’s test for multiple testing was used to obtain multiple testing of *p* values. Statistical significance: *, *p* < 0.05; **, *p* < 0.01; ***, *p* < 0.001; ****, *p* < 0.0001.

**Figure 2 microorganisms-09-01742-f002:**
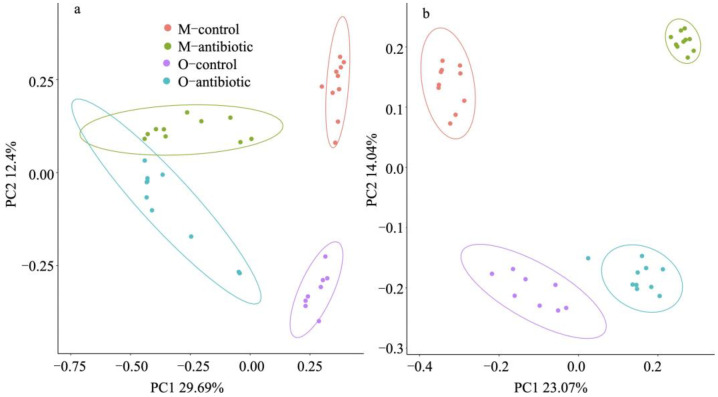
PCoA based on Bray–Curtis dissimilarities (**a**) and unweighted UniFrac (**b**) of the gut microbiota in the maternal and offspring samples. M-control and M-antibiotic stand for control mothers and antibiotics-treated mothers, respectively, while O-control and O-antibiotic represent offspring born to control and antibiotics-treated mothers, respectively. The model of mouse type × treatment was used, whereby mouse type is mother or offspring and treatment is antibiotic or control. Adonis: R^2^ = 0.484, *p* = 0.001; R^2^ = 0.425, *p* = 0.001. Presence of ellipses is indicative of a 95% confidence interval of β-diversity measures.

**Figure 3 microorganisms-09-01742-f003:**
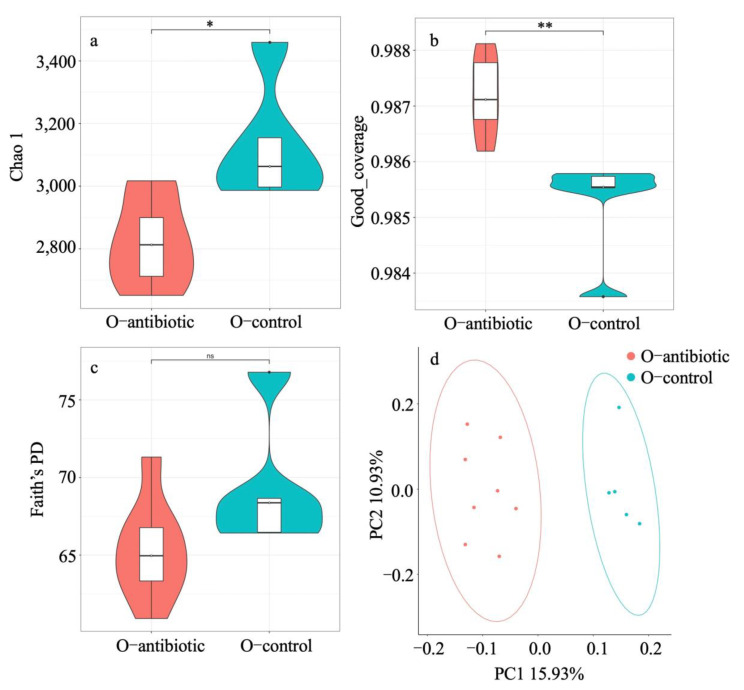
Comparison of the Chao 1 (**a**), good_coverage (**b**) and Faith’s PD (**c**) and principal-coordinates analysis (PCoA) based on unweighted UniFrac distance matrix (**d**) of 14-week old offspring born to antibiotics-treated mothers and controls. O-antibiotic and O-control stand for offspring born to antibiotics-treated mothers and controls, respectively. Statistical significance: *, *p* < 0.05; **, *p* < 0.01; ns, not significant. Presence of ellipses is indicative of a 95% confidence interval of β-diversity measures.

**Figure 4 microorganisms-09-01742-f004:**
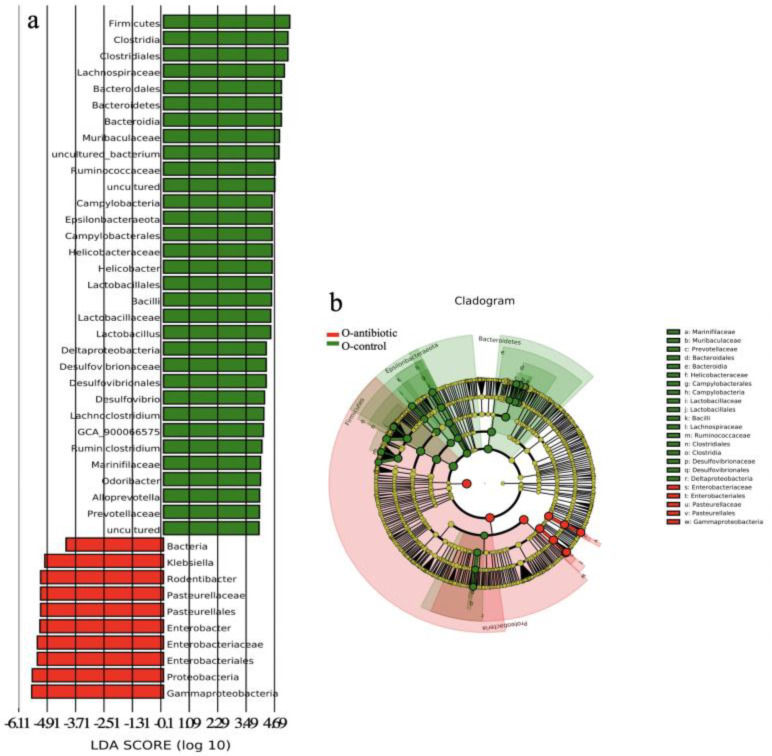
The LDA (linear discriminant analysis) score (**a**) and the taxonomic cladogram (**b**) obtained from linear discriminant analysis effect size (LEfSe) analysis of the gut microbiota of offspring born to antibiotics-treated mothers and controls. Bar chart showing the log-transformed LDA scores of bacterial taxa identified by LEfSe analysis (the log-transformed LDA score of 5.5 as the threshold). Cladogram showing the phylogenetic relationships of bacterial taxa revealed by LEfSe. From inside to outside, the circle of radiation represented the classification level from phylum to genus. O-antibiotic and O-control represent offspring born to antibiotics-treated mothers and controls, respectively.

**Figure 5 microorganisms-09-01742-f005:**
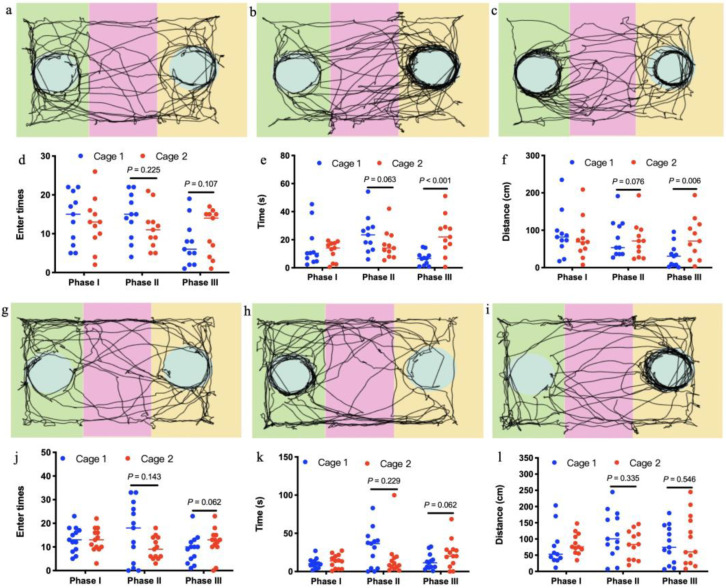
Social interactions of offspring mice born to control mothers (**a**–**f**, *n* = 11) and those treated with antibiotics (**g**,**k**, *n* = 13). In the three-chambered sociability test, offspring born to mothers treated with antibiotics showed reduced sociability and preference for social novelty displayed by those from controls, as seen in the automated tracking images at phase I (**a**,**g**), phase II (**b**,**h**) and phase III (**c**,**i**), the entry times in each chamber at phase I, II, and III (**d**,**j**), the time spent sniffing in each chamber at phase I, II, and III (**e**,**k**) and the travel distances in each chamber at phase I, II, and III (**f**,**l**).

**Figure 6 microorganisms-09-01742-f006:**
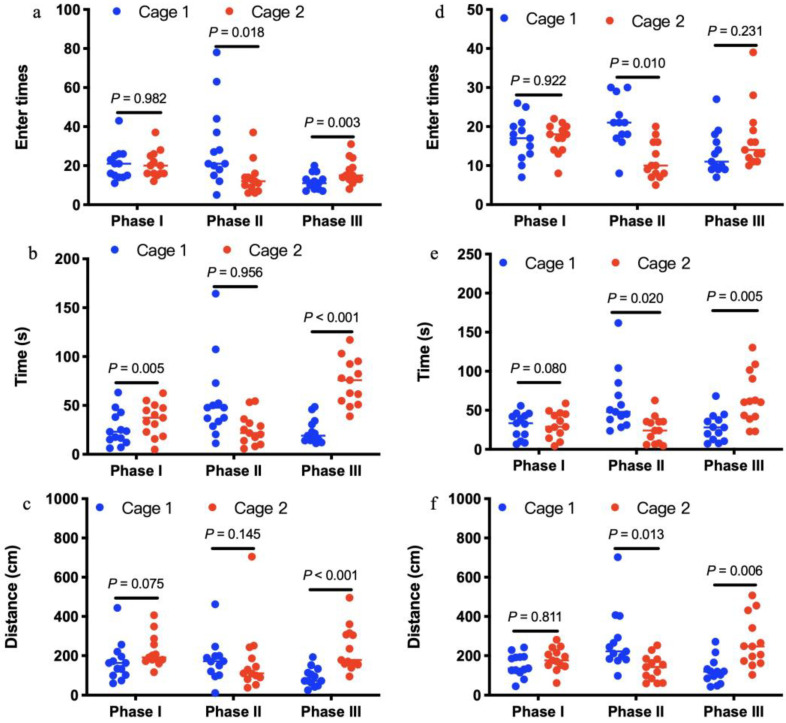
Social interactions of antibiotics-treated mothers (**a**–**c**, *n* = 13) and controls (**d**–**f**, *n* = 13). In the three-chambered sociability test, antibiotics-treated mothers exhibited similar sociability and preference for social novelty to controls, as seen from the entry times (**a**,**d**), the time spent sniffing (b and e), and the travel distances (**c**,**f**) in each chamber at phase I, II, and III.

## Data Availability

Data will be deposited in the European Nucleotide Archive upon acceptance for publication.

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
