# Peer review of "Maternal Gut Dysbiosis Alters Offspring Microbiota and Social Interactions"

_microorganisms, 2021, doi:10.3390/microorganisms9081742_

Round 1

Reviewer 1 Report

All of my concerns have been addressed.

Reviewer 2 Report

I don't have any further notes

This manuscript is a resubmission of an earlier submission. The following is a list of the peer review reports and author responses from that submission.

Round 1

Reviewer 1 Report

Ju et al add support to the existing literature that early life microbiota affects behavior. Their work is well constructed and well powered. Presentation of the work and some analyses need improvement.

Major comments

Lines 386 to 388: With regards to early life microbiota alterations altering behavior, please expand your discussion. Consider the following references and others to support your observations. Notably these put forth mechanisms by which the early life microbiota influences enteric neuron and brain development as well as behavior: PMIDs: 32390463; 30522820

Add citations of R, ggplot2, and other R packages used for plots and analyses.

Add a diagram of the study design, when giving antibiotics, when removing animals for sequencing, when do testing, when weaning to individual cages, etc.

Lines 205 to 214: Are these data rarefied? I would recommend removing this whole paragraph as your subsequent data are much more robust and better presented. If you wish to highlight pairwise comparisons of sample groups, use boxplots of distance metric distances between and within groups. For example: http://qiime.org/tutorials/creating_distance_comparison_plots.html

Fig 1a-d. What stat test with used? An ANOVA with correction for multiple testing is recommended to obtain pair-wise pvalues. Panels need to be labeled. X axis label for top panels are cut off. Also, add Faith’s PD for a phylogenetic look of alpha diversity.

Fig 1 legend: line 235 to 237 should read “M-control and M-antibiotics stand for control mothers and antibiotics-treated mothers, respectively while O-control and O-antibiotic represent offspring born to control and antibiotic-treated mothers, respectively”. Same problem in Figs 2, 3, 4, 5,6.

Fig 2: Present F-stat, R2, and pvalue from the PERMANOVA (adonis) output and put them in the fig legend or on the figure. Describe the model given used for the PERMANOVA. Your model should be mousetype*treatment, whereby mouse type is mother or offspring and treatment is antibiotic or control. What did you use to make the circles on the plot? Consider from switching Bray-Curtis to weighted or unweighted Unifrac for the distance metrics. A phylogenetic analysis will give better insight into your data.

Line 222: the statement might be better supported with distance comparison boxplots as described above.

Fig S2: Can you use the same abbreviations as in Fig 2. What alpha diversity metrics are in A and B? What beta-diversity analysis in C? Why not combine these with Figs 1 and 2?

Line 246: why as expected?

FigS3: order as M-control, M-antibiotic, O-control, O-antibiotic

Lines 244-258: what stat test was used? And what multiple testing correction? List next to pvalue

Fig 4: quality is low

Fig 5: very low quality

Fig 7: cannot read, especially the legend.

Data need to be deposited in an archive like European Nucleotide Archive before acceptance. Figshare is not for microbiome data.

Why include the LefSe analysis as well as the analysis in lines 244-258? Please compress these sections. The presentation is excessive.

Why focus on the sequencing results from younger pups and not discuss the old pups? Please explain your rationale.

Line290 to 313: confused. Metabolite analysis was done according to the methods. How were the metabolites extracted? Is that what the KEGG data is analyzing. Please clarify the methods and text. Are these data from PiCRUST? What stats were done? The biggest difference in Fig 6 seems to be the clustering of the O-antibiotic. This section may need to be removed. As written it is very unclear in design, purpose, and results.

Paper needs to remove figures and explain the flow of their experiments better.

Fig 8 and S4: These plots are not interpretable. Label which is control and antibiotic treated. Add pvaules and what stat was used. Show all of the points and say the N per group in the figure legend.

S4 is an important finding in the interpretation of your results. Move to main text.

Overall, I would recommend shortening the paper, removing lines 205-214, compressing 244 to 268, removing 290 to 313, and removing figures 6 and 7, S1, moving figures 3 and 4 to supplement, combining S2 with Figs 1 and 2, and moving S4 to main test.

Minor comments

Line 77 give definition of KM

Lines 88, 89, and 91: pubs -> pups

Why does line 92 start with “however”?

Line 215: Many other groups have shown effects of antibiotics on the microbiome. Your findings are not novel. Please adjust the phrasing of this line.

Line 282: remove the “and” at the end of the line

Fig 3&4: Epsilonbacteraeota spelling line

Author Response

Lines 386 to 388: With regards to early life microbiota alterations altering behavior, please expand your discussion. Consider the following references and others to support your observations. Notably these put forth mechanisms by which the early life microbiota influences enteric neuron and brain development as well as behavior: PMIDs: 32390463; 30522820

We expanded our discussion and cited the suggested refs.

Add citations of R, ggplot2, and other R packages used for plots and analyses.

We added in the res list as suggested.

Add a diagram of the study design, when giving antibiotics, when removing animals for sequencing, when do testing, when weaning to individual cages, etc.

We added more information in the Method section and made it clearer. Therefore, we think a diagram of the study design is not necessary. Hope the reviewer agrees.

Lines 205 to 214: Are these data rarefied? I would recommend removing this whole paragraph as your subsequent data are much more robust and better presented. If you wish to highlight pairwise comparisons of sample groups, use boxplots of distance metric distances between and within groups. For example: http://qiime.org/tutorials/creating_distance_comparison_plots.html

We removed the whole para as suggested.

Fig 1a-d. What stat test with used? An ANOVA with correction for multiple testing is recommended to obtain pair-wise pvalues. Panels need to be labeled. X axis label for top panels are cut off. Also, add Faith’s PD for a phylogenetic look of alpha diversity.

We done these as suggested.

Fig 1 legend: line 235 to 237 should read “M-control and M-antibiotics stand for control mothers and antibiotics-treated mothers, respectively while O-control and O-antibiotic represent offspring born to control and antibiotic-treated mothers, respectively”. Same problem in Figs 2, 3, 4, 5,6.

We changed throughout the text as suggested.

Fig 2: Present F-stat, R2, and pvalue from the PERMANOVA (adonis) output and put them in the fig legend or on the figure. Describe the model given used for the PERMANOVA. Your model should be mousetype*treatment, whereby mouse type is mother or offspring and treatment is antibiotic or control. What did you use to make the circles on the plot? Consider from switching Bray-Curtis to weighted or unweighted Unifrac for the distance metrics. A phylogenetic analysis will give better insight into your data.

We added F-stat, R2, and pvalue. Circles were generated to well illustrate the difference between treatments. We switched Bray-Curtis to unweighted Unifrac for the distance metrics. In addition, a phylogenetic analysis was given.

Line 222: the statement might be better supported with distance comparison boxplots as described above.

Yes, we changed as suggested.

Fig S2: Can you use the same abbreviations as in Fig 2. What alpha diversity metrics are in A and B? What beta-diversity analysis in C? Why not combine these with Figs 1 and 2?

We used the same abbreviations as suggested and made corrections to the figures. We also moved Fig. S2 into the main text as Fig. 3.

Line 246: why as expected?

Removed

FigS3: order as M-control, M-antibiotic, O-control, O-antibiotic

We changed as suggested.

Lines 244-258: what stat test was used? And what multiple testing correction? List next to pvalue

General linear model was used. We added this information in Method section as suggested.

Fig 4: quality is low

We changed as suggested.

Fig 5: very low quality

We changed as suggested.

Fig 7: cannot read, especially the legend.

We changed as suggested.

Data need to be deposited in an archive like European Nucleotide Archive before acceptance. Figshare is not for microbiome data.

We will deposit our data in European Nucleotide Archive upon the final acceptance.

Why include the LefSe analysis as well as the analysis in lines 244-258? Please compress these sections. The presentation is excessive.

LefSe analysis will further help us detect the differences. However, we compressed these sections as suggested.

Line290 to 313: confused. Metabolite analysis was done according to the methods. How were the metabolites extracted? Is that what the KEGG data is analyzing. Please clarify the methods and text. Are these data from PiCRUST? What stats were done? The biggest difference in Fig 6 seems to be the clustering of the O-antibiotic. This section may need to be removed. As written it is very unclear in design, purpose, and results.

We removed these sections as suggested.

Paper needs to remove figures and explain the flow of their experiments better.

Fig 8 and S4: These plots are not interpretable. Label which is control and antibiotic treated. Add pvaules and what stat was used. Show all of the points and say the N per group in the figure legend.

Changed as suggested

S4 is an important finding in the interpretation of your results. Move to main text.

Moved as suggested

Overall, I would recommend shortening the paper, removing lines 205-214, compressing 244 to 268, removing 290 to 313, and removing figures 6 and 7, S1, moving figures 3 and 4 to supplement, combining S2 with Figs 1 and 2, and moving S4 to main test.

All were changed as suggested

Minor comments

Line 77 give definition of KM

Changed as suggested

Lines 88, 89, and 91: pubs -> pups

Changed as suggested

Why does line 92 start with “however”?

Removed as suggested

Line 215: Many other groups have shown effects of antibiotics on the microbiome. Your findings are not novel. Please adjust the phrasing of this line.

Changed “significant”as suggested

Line 282: remove the “and” at the end of the line

Changed as suggested

Fig 3&4: Epsilonbacteraeota spelling line

Changed as suggested

Reviewer 2 Report

The manuscript presented for review deserves interest as a well-written text and thrilling story. Questions raised in an article are currently of a high value and insights of alterations in maternal gut dysbiosis in its relationship with offspring microbiota, metabolome and social interactions could play vital role in further understanding of the problematique.

This manuscript is definitely worth consideration for acceptation after some minor corrections of the text:

  1. Proof read of the text for typos and small edits such as:
    1. L77 – please explain abbreviation KM
    2. L195 – please correct typo “metabonomics”
  2. Having a miscellaneous list of references, this manuscript still can benefit from including several recent studies on the topic of microbiome alterations https://doi.org/10.3390/microorganisms8081225 (2020)
    https://doi.org/10.14814/phy2.14610 (2020)
    https://doi.org/10.1186/s12866-021-02099-0 (2021)

 Otherwise, I would like to greet authors with a nice written manuscript and wish them further success.

Author Response

    • Proof read of the text for typos and small edits such as:
      1. L77 – please explain abbreviation KM
    • Changed as suggested
      1. L195 – please correct typo “metabonomics”
    • Changed as suggested
    • Having a miscellaneous list of references, this manuscript still can benefit from including several recent studies on the topic of microbiome alterations https://doi.org/10.3390/microorganisms8081225 (2020)
      https://doi.org/10.14814/phy2.14610 (2020)
    • Added as suggested

Round 2

Reviewer 1 Report

Zhang et al have improved their manuscript but numerous places are still not of publishable quality. Importantly, the main sociability figures have not been improved and may contain some errors.

Line 14 and 17: metabolome is still mentioned in the abstract despite these data not being present in the manuscript

Line 34: “contributing negative impacts to the healthcare and humanity” -> “contributing negative impacts to the healthcare of humanity”

Line 181: What is the GLM used?

Figure 2: Only one PERMANOVA (Adonis) stat is shown while there are two different panels. Please provide PERMANOVA stat for both or at least specify for which panel the shown stats refer to. Same with line 190.

Line 196: This wording suggests the Anosim was only performed on the data in panel B, the unweighted Unifrac. Please clarify Fig 2 legend and cite Fig 2b instead of Fig 2 in line 196.

Line 192: “Significantly reduction of” -> ““Significantly reduced”

Fig 3c isn’t referred in the text

Line 202: Why is Fig 3d labeled as additional file?

Legends for figures S1 through S3 still have the definitions of M-control and M-antibiotics and O-control and O-antibiotics flipped.

Panels for Fig S3 are not labeled though they are referred in the text at line 246.

There are two Fig 4s.

Fig 4 (LefSe): quality is bad. The x-axis of panel a and part of the cladogram in panel b is cut off with a legend. Also the definitions of O-control and O-antibiotics are flipped

Fig 4 (sociability) and fig 5 need to show individual points and pvalues on the figure and give the N per group and stats method used in the legend. For the automated tracking images, which phase is shown or are all three phase shown at once? Please clarify the legend.

Lines 262 to 285: label which panel in the figure the stats data are from

Line 268 to 270: The pvalue here is 0.107, which is not significant. Please adjust the wording of these lines.

Line 275 to 277: What phase is being discussed here.

Line 281: a pvalue of 0.062 here is not considered slightly significant but in line 266, a pvalue of 0.063 is considered marginal. Please be consistent with interpretations of the data.

Fig 4 (sociability): If I am reading the text correctly, the pvalue for panel c, phase III is p<0.001 and that for panel d, phase III is p=0.006. With the plots shown, it is very difficult to see how those pvalues could be true. If anything panel g phase III looks significant and maybe panel h phase III. Is there incorrect labeling of panels b-d and f-h? Please check if b-d are actually the antibiotic treated and f-h are the controls and check that the correct pvalues are reported.

Line 285: provide pvalues for Fig 5 data.

Figure 5: as noted earlier, show individual points, pvalues and explain the stats used in the figure legend. The reader cannot interpret the data as is. Also why are only enter times shown? It is important to show that for sniffing and distance since those differences were reported as different between antibiotic and control offspring in lines 271 to 275.

Citation 50 lists the first names not that the last names in full.

Author Response

Line 14 and 17: metabolome is still mentioned in the abstract despite these data not being present in the manuscript

Deleted

Line 34: “contributing negative impacts to the healthcare and humanity” -> “contributing negative impacts to the healthcare of humanity”

Changed, see line 33

Line 181: What is the GLM used?

It stands for General linear model, see line 180

Figure 2: Only one PERMANOVA (Adonis) stat is shown while there are two different panels. Please provide PERMANOVA stat for both or at least specify for which panel the shown stats refer to. Same with line 190.

Changed, see line 189-190

Line 196: This wording suggests the Anosim was only performed on the data in panel B, the unweighted Unifrac. Please clarify Fig 2 legend and cite Fig 2b instead of Fig 2 in line 196.

Changed, see line 195

Line 192: “Significantly reduction of” -> ““Significantly reduced”

Changed, see line 191

Fig 3c isn’t referred in the text

Added, see line 198

Line 202: Why is Fig 3d labeled as additional file?

Deleted, see line 201

Legends for figures S1 through S3 still have the definitions of M-control and M-antibiotics and O-control and O-antibiotics flipped.

Changed, see figures

Panels for Fig S3 are not labeled though they are referred in the text at line 246.

Deleted the sentence.

There are two Fig 4s.

Changed as suggested.

Fig 4 (LefSe): quality is bad. The x-axis of panel a and part of the cladogram in panel b is cut off with a legend. Also the definitions of O-control and O-antibiotics are flipped

Changed as suggested.

Fig 4 (sociability) and fig 5 need to show individual points and pvalues on the figure and give the N per group and stats method used in the legend. For the automated tracking images, which phase is shown or are all three phase shown at once? Please clarify the legend.

Changed as suggested.

Lines 262 to 285: label which panel in the figure the stats data are from

Added in the whole para as suggested.

Line 268 to 270: The pvalue here is 0.107, which is not significant. Please adjust the wording of these lines.

Changed as suggested, see line 265.

Line 275 to 277: What phase is being discussed here.

Added as suggested, see line 273.

Line 281: a pvalue of 0.062 here is not considered slightly significant but in line 266, a pvalue of 0.063 is considered marginal. Please be consistent with interpretations of the data.

Changed as suggested, see line 262-263.

Fig 4 (sociability): If I am reading the text correctly, the pvalue for panel c, phase III is p<0.001 and that for panel d, phase III is p=0.006. With the plots shown, it is very difficult to see how those pvalues could be true. If anything panel g phase III looks significant and maybe panel h phase III. Is there incorrect labeling of panels b-d and f-h? Please check if b-d are actually the antibiotic treated and f-h are the controls and check that the correct pvalues are reported.

The reviewer is correct. We changed as suggested.

Line 285: provide pvalues for Fig 5 data.

Provided, see line 281-287

Figure 5: as noted earlier, show individual points, pvalues and explain the stats used in the figure legend. The reader cannot interpret the data as is. Also why are only enter times shown? It is important to show that for sniffing and distance since those differences were reported as different between antibiotic and control offspring in lines 271 to 275.

Added as suggested in Fig 6.

Citation 50 lists the first names not that the last names in full.

Changed, see line 489

Round 3

Reviewer 1 Report

I thank the authors for greatly improving and fine tuning their manuscript. I have a couple of minor comments, but otherwise the manuscript has addressed my concerns.

There are two Fig 5s now, the second one should become Fig 6.

I would still like to see stars or pvalues on Figs 5 and 6 so that a reader can simply look at the figure rather than read the text to find the pvalues.

Thank you for your efforts and science.

Author Response

There are two Fig 5s now, the second one should become Fig 6.

Our response: We changed another Fig. 5 into Fig. 6.

I would still like to see stars or pvalues on Figs 5 and 6 so that a reader can simply look at the figure rather than read the text to find the pvalues.

Our response: We added p values on Fig 5 and Fig 6, but removed them from the text to avoid repetition.